# Modification of Dispersin B with Cyclodextrin-Ciprofloxacin Derivatives for Treating Staphylococcal

**DOI:** 10.3390/molecules28145311

**Published:** 2023-07-10

**Authors:** Jinan Abdelkader, Magbool Alelyani, Yazeed Alashban, Sami A. Alghamdi, Youssef Bakkour

**Affiliations:** 1Laboratory of Applied Chemistry (LAC), Department of Chemistry, Faculty of Sciences III, Lebanese University Mont Michel, El Koura 826, Lebanon; 2Department of Radiological Sciences, College of Applied Medical Science, King Khalid University, Abha 61421, Saudi Arabia; 3Radiological Sciences Department, College of Applied Medical Sciences, King Saud University, P.O. Box 145111, Riyadh 4545, Saudi Arabia

**Keywords:** cyclodextrin, ciprofloxacin, Dispersin B, inclusion complex, antibiofilm

## Abstract

To address the high tolerance of biofilms to antibiotics, it is urgent to develop new strategies to fight against these bacterial consortia. An innovative antibiofilm nanovector drug delivery system, consisting of Dispersin B-permethylated-β-cyclodextrin/ciprofloxacin adamantyl (DspB-β-CD/CIP-Ad), is described here. For this purpose, complexation assays between CIP-Ad and (i) unmodified β-CD and (ii) different derivatives of β-CD, which are 2,3-O-dimethyl-β-CD, 2,6-O-dimethyl-β-CD, and 2,3,6-O-trimethyl-β-CD, were tested. A stoichiometry of 1/1 was obtained for the β-CD/CIP-Ad complex by NMR analysis. Isothermal Titration Calorimetry (ITC) experiments were carried out to determine Ka, ΔH, and ΔS thermodynamic parameters of the complex between β-CD and its different derivatives in the presence of CIP-Ad. A stoichiometry of 1/1 for β-CD/CIP-Ad complexes was confirmed with variable affinity according to the type of methylation. A phase solubility study showed increased CIP-Ad solubility with CD concentration, pointing out complex formation. The evaluation of the antibacterial activity of CIP-Ad and the 2,3-O-dimethyl-β-CD/CIP-Ad or 2,3,6-O-trimethyl-β-CD/CIP-Ad complexes was performed on *Staphylococcus epidermidis* (*S. epidermidis*) strains. The Minimum Inhibitory Concentration (MIC) studies showed that the complex of CIP-Ad and 2,3-O-dimethyl-β-CD exhibited a similar antimicrobial activity to CIP-Ad alone, while the interaction with 2,3,6-O-trimethyl-β-CD increased MIC values. Antimicrobial assays on S. epidermidis biofilms demonstrated that the synergistic effect observed with the DspB/CIP association was partly maintained with the 2,3-O-dimethyl-β-CDs/CIP-Ad complex. To obtain this “all-in-one” drug delivery system, able to destroy the biofilm matrix and release the antibiotic simultaneously, we covalently grafted DspB on three carboxylic permethylated CD derivatives with different-length spacer arms. The strategy was validated by demonstrating that a DspB-permethylated-β-CD/ciprofloxacin-Ad system exhibited efficient antibiofilm activity.

## 1. Introduction

*S. epidermidis* is a major infective agent in compromised patients, such as drug abusers or immunocompromised patients (for example, patients under immunosuppressive therapy, chronic wound patients, AIDS patients, and premature newborns). The ability to adhere and subsequently form biofilms on indwelling devices is among the potential virulence factors associated with *S. epidermidis* species [1,2,3]. Indeed, sessile organisms exhibit a high resistance to antimicrobials compared to their planktonic counterparts [4].

The Extracellular Polymeric Substances (EPSs) of the biofilm matrix form the scaffold for a three-dimensional architecture, being involved in bacterial adhesion and biofilm cohesion [5]. The EPS matrix also contributes to bacterial resistance against nonspecific and specific host defenses and antimicrobial agents [6,7,8,9,10,11,12]. Polysaccharides are a major fraction of this EPS matrix [5,13]. Most of them are linear or branched long molecules. Some of these polysaccharides are homopolysaccharides, including sucrose-derived glucans and fructans [13], but most of them are heteropolysaccharides that consist of a mixture of neutral and charged sugar residues. They can contain organic or inorganic substitutions that significantly affect their physical and biological properties.

Polycationic exopolysaccharides also exist, such as intercellular adhesin, which is composed of β-1,6-linked N-acetylglucosamine with partly deacetylated residues. The matrix-mediated antibiotic resistance is mainly due to a low metabolic activity and oxygen limitation within the biofilm [14]. The role of reduced antibiotic penetration in the drug resistance of biofilms is highly antibiotic-dependent, with beta-lactams and vancomycin diffusion being significantly reduced, whereas the diffusion of aminosides and fluoroquinolones seems unaffected [14,15,16]. Some efficient strategies for biofilm eradication are combinations of bacteria killing and matrix removal by using Detachment-Promoting Agents (DPAs). Many of these DPAs have been described in the literature, including enzymes and chemical agents [17]. They include chelating agents, for example, EGTA [18], EDTA, NaCl, CaCl_2_, or MgCl_2_ [17]. Among agents having a promising clinical future are biofilm-matrix-degrading enzymes. Indeed, numerous studies have pointed out the ability of these enzymes to inhibit biofilm proliferation, detach preformed biofilms, and sensitize biofilms toward antimicrobials by depolymerizing either polysaccharides [19,20] or extracellular DNA [21]. DispersinB^®^ (DspB) [22] was discovered in 2003 from Actinobacillus actinomycetemcomitans. Treatment of S. epidermidis biofilms with DspB caused the dissolution of the EPS matrix and detachment of biofilm cells from the surface [20,23], and disrupted biofilm formation by *Escherichia coli*, *S. epidermidis*, *Yersina pestis*, and *Pseudomonas fluorescens* [19]. One of the main drawbacks of the enzyme-based anti-biofilm strategy is that the dispersal of bacteria from the biofilm may favor bloodstream infections, septic thrombophlebitis, endocarditis, metastatic infections, and sepsis [2,24,25,26,27]. It is the reason why, most of the time, it is simultaneously or sequentially used in combination with antimicrobial agents.

Several studies have also showed that DspB sensitizes biofilm bacteria to antibiotics [28,29,30] and macrophages [31]. A preliminary experiment, performed here, confirmed that DspB enhanced the activity of the ciprofloxacin (CIP) fluoroquinolone against biofilms when used simultaneously.

Synthetic fluoroquinolones and derivatives are broad-spectrum antibiotics that are widely used for the treatment of serious bacterial infections for human and veterinary use [32,33]. The structure–activity relationship of fluoroquinolones has been extensively investigated and depends on the presence of the carboxyl, carbonyl, and fluorine groups at C_3_, C_4_, and C_6_ positions, respectively [34]. Moreover, the antibacterial effectiveness improves significantly if an amino substituent is present at C_5_, an alkylated pyrrolidine or piperazine at C_7_, a halogen at C_8_, and a cyclopropyl group at N1 [34]. An interesting approach is the development of new quinolone-based antibacterial agents by chemical modification on the carboxylic acid for the preparation of quinolone–macrocycle conjugates [35]. Ciprofloxacin is one of the most used fluoroquinolones in the clinical field with, for example, the treatment of respiratory infections [36,37], meningitis [36], septicemia [36], and intra-abdominal infections [38]. The aim of our study was to design a new antibiofilm vector drug delivery system, consisting of a DspB-β-cyclodextrin (β-CD)/CIP-Ad complex, combining both enzyme and antibiotic delivery. This vehicle is an “all-in-one” tool, which provides the ability to simultaneously destroy the biofilm matrix and release the antibiotic to achieve a combined effect (Figure 1).

β-CDs are truncated cone-shaped molecules composed of seven α(1-4)-glucose units. The 7 primary alcohol functions at the C-6 position and the 14 secondary alcohols at the C-2 and C-3 positions constitute the primary and secondary faces of the macrocycle, respectively. This conformation forms a hydrophobic central cavity suitable for the inclusion of various organic molecules and a hydrophilic external surface. The main characteristic of CDs is the ability to insert the hydrophobic part of a guest molecule into the cavity. The formation of this host–guest inclusion complex and its stability also depend on the size and functionalization of the molecule [39].

The supramolecular chemistry of CDs has often been used in enzyme technology [40] with, for illustration, the works of Villalonga et al. [41,42,43,44,45]. The authors demonstrated that the immobilization of CD on the surface of enzymes improved its affinity for the substrate and its thermostability while maintaining a good activity.

In order to overcome the impediment due to this weak interaction between the guest antibiotic and the host CD molecules, we synthesized an adamantyl ciprofloxacin derivative (CIP-Ad). Indeed, the high affinity of the adamantyl group for the internal cavity of the macrocycle [46] will improve the stability of the inclusion complex with the ciprofloxacin. Moreover, such a supramolecular interaction should enhance the antibiotic solubility, stability, and bioavailability [47,48]. To develop this new concept, we first had to form an inclusion complex between the modified antibiotic (CIP-Ad) and various β-CDs, by preserving the antimicrobial activity.

Complexation of the CIP-Ad with native β-CD I and three commercially available methylated β-CD derivatives, i.e., 2,3-O-dimethyl-β-CD II, 2,6-O-dimethyl-β-CD III, and 2,3,6-O-trimethyl-β-CD IV (Figure 2), was thus performed, and investigated using isothermal titration calorimetry (ITC), in order to evaluate thermodynamic parameters of the interactions. The stoichiometry of the obtained complexes was determined using two-dimensional Nuclear Overhauser Effect Spectroscopy (NOESY). The antibiofilm efficacy of the DspB-CIP and DspB (2,3-O-dimethyl-β-CDs/CIP-Ad) complex was then tested on a mature *S. epidermidis* biofilm model. To reach the one-shot strategy, CDs were grafted on DspB via a covalent coupling strategy. In this context, we synthesized three carboxylic permethylated CD derivatives with different spacer arm lengths at the C-6 position of CDs I, II, and III (Figure 2). The antibiofilm activity of DspB-CD conjugates was then investigated to evaluate the influence of such a modification on the biofilm matrix.

Finally, we validated our approach by demonstrating that the DspB-2,3,6-O-trimethyl-β-CD/CIP-Ad drug delivery system exhibits an efficient antibiofilm activity.

## 2. Results

The use of an adamantyl group was chosen as a molecular anchor to enhance the intermolecular interactions between the enzyme-conjugated CD and the antibiotic [46]. Furthermore, it was shown that adamantaplatensimycin (bioactive analog of platensimycin) conserves the antibacterial activity against methicillin-resistant *Staphylococcus aureus* and vancomycin-resistant *Enterococcus faecium* [49]. The synthesis of the CIP-Ad 4 derivative was obtained in three steps.

(Figure 1). After the quantitative amino protection step, an efficient peptide coupling reagent was used to overcome the low reactivity of C-3 carboxylic acid of the quinolone core. After the deprotection step, the desired CIP-Ad 4 was obtained quantitatively.

### 2.1. NMR Study

The aim of the NMR study was to highlight and describe the inclusion properties and factors affecting the complexation between CIP-Ad 4 and β-CD I. In the structure of β-CDs, H_3_ and H_5_ protons are located inside the cavity, whereas H_2_ and H_4_ are outside the torus (Figure 3). The H6 protons of the primary alcohol group are on the narrow side, and the H_1_ protons are in the glycosidic bond plane of β-CD. Therefore, the formation of inclusion complexes is usually highlighted, in the proton NMR spectrum, by the shift and the deformation of H_3_ and H_5_ NMR signals.

So, an assignment of ^1^H NMR spectra of the free forms of CIP-Ad 4, β-CD I, and the β-CD I/CIPAd 4 complex in D2O/DCl at pH = 3 (Figure 3) was achieved by 2D NMR 1H/1H COSY (Figure 1) as well as ^1^H/^1^H NOESY (Figure 2), and the chemical shifts are summarized in Table 1. As expected, β-CD’s H_3_ and H_5_ proton signals were the most affected by the inclusion as their signals were significantly broadened and shifted by Δδ = 0.02 ppm and 0.01 ppm, respectively. Regarding the CIP-Ad 4 part, the chemical shifts of adamantyl protons H_11_, H_11′_, and H_12_ and the aromatic protons H_2_, H_5_, and H_8_ were modified by the inclusion.

The classic method of Job’s Plot, usually used to determine the stoichiometry of this type of complex, was not effective in our case, because of the very low solubility of the CIP-Ad **4** molecule. Nevertheless, from the structural information obtained from the 2D ^1^H/^1^H NOESY spectrum of the complex, it was possible to determine the orientation of the CIP-Ad **4** molecule in the cavity of the β-CD and to indirectly deduce the stoichiometry of the β-CD **I**/CIP-Ad **4** complex. The 2D ^1^H/^1^H NOESY spectrum of the β-CD **I**/CIP-Ad **4** complex (Figure 4) exhibited cross peaks between the adamantyl protons (H11, H11′, H12) of CIP-Ad **4** and protons H3 and H5 of β-CD **I,** confirming the inclusion of the adamantyl group of CIP-Ad **4** in the β-CD cavity. On the contrary, the absence of correlation between H3 and H5 of β-CD **I** and H7a and H7b of CIP-Ad **4** suggested no inclusion of the piperazinyl group.

These results were in favor of a β-CD I/CIP-Ad 4 complex with a 1/1 stoichiometry by the adamantyl side of CIP-Ad **4** (Figure 5).

### 2.2. ITC Studies

ITC experiments were then carried out for the determination of thermodynamic parameters (Ka, ΔH, ΔS) and stoichiometries of CIP-Ad and β-CD I, 2,3-*O*-dimethyl-β-CD II, 2,6-*O*-dimethyl-β-CD III, and 2,3,6-*O*-trimethyl-β-CD IV complexes (Figure 2). Experiments were performed in acidic medium, leading to the protonated form of CIP-Ad 4. The heat generated during the association process was provided in μJ/mol of titrant (host, I, II, III, IV), in respect of the titrant/titrate molar fraction (host, I, II, III, IV/guest CIP-Ad). The experimental titration curve was fitted through a one-site-independent binding model [52]. The value of the apparent binding stoichiometry (n) was extracted by the inflection point of the curve. As all the curves were perfectly sigmoidal, an accurate determination of n was obtained.

The thermodynamic parameters were determined at different temperatures (15–60 °C) for all β-CD (I, II, III, IV) hosts. The experimental titration curves of CIP-Ad 4 by β-CD I at 25 °C showed an equivalence point for approximately a 1/1 molar ratio of β-CD/CIP-Ad, confirming the observed NMR data. The same stoichiometry was obtained for all host molecules (II, III, IV). The fitting of the binding curves gave direct measurements of affinity constant Ka and enthalpy variation ΔH; then, thanks to thermodynamic equations, Gibbs free energy ΔG and entropy variation ΔS (Table 2) were determined [52]. The variation in heat capacity ΔCp values was calculated as the slope of the curve correlated with the enthalpy values versus the temperature.

Usually, to validate the ITC experiments, Wiseman’s “C” parameter [53] has to be between 10 and 500–1000 [53,54]. In the present study, all C values are in the optimal experimental window, except for the II/CIP-Ad complex (C value around 2–3) (Table 2). However, Turnbull and Daranas demonstrated that even for a low-affinity system with a weak C value from 0.01 to 10 (Ka = 10^2^ M^−1^), the ITC results are still valid [55]. In this case, three conditions must be respected: (*i*) use of a sufficient portion of the binding isotherm for analysis; (*ii*) precise knowledge of the binding stoichiometry n and of the concentrations of ligand and receptor; (*iii*) an adequate signal-to-noise ratio. These criteria are filled for the present study with the CIP-Ad/β-CD II complex. The host–guest inclusion complex represents the distribution of species in equilibrium giving an average stoichiometry and a binding constant Ka [39,52]. In the present study, this equilibrium constant Ka was calculated for the CIP-Ad/β-CD I complex of around 10^5^ M^−1^ at 37 °C (i.e., ΔG ≈ −30 kJ/mol). This strong association constant is due to the perfect fitting of the adamantyl moiety inside the β-CD cavity. Consequently, the presence of a large CIP structure bound to the adamantyl group produced no change in the inclusion process, as reported in the literature with other adamantane derivatives (ΔG° ≈ −20 to −30 kJ/mol) [56,57]. Different values of the association constant were obtained for methylated β-CDs (II, III, and IV) ranging from 10^4^ to 10^6^ M^−1^ at 37 °C (i.e., ΔG from −23 to −35 kJ/mol). The highest binding constant was observed with the CIP-Ad/β-CD III (Ka = 72.7 10^4^ M^−1^), followed by the CIP-Ad/β-CD I (Ka = 21.3 10^4^ M^−1^), CIP-Ad/β-CD IV (Ka = 8.4 10^4^ M^−1^), and CIP-Ad/β-CD II (Ka = 1.0 10^4^ M^−1^) complexes. The binding strength is clearly dependent on the host molecule functionalization. Such differences have been already reported in the literature with adamantyl derivatives in favor of β-CD [56,57]. We observed a spontaneous complexation (ΔG < 0) for all CIP-Ad/β-CD complexes (I to IV) with thermodynamically favorable enthalpy and entropy values (ΔH < 0 and ΔS > 0, respectively). Complexations of CIP-Ad with I and IV are enthalpy-driven (|ΔH| > |TΔS|) on the full temperature range studied, while interactions with II and III are entropy-driven (|ΔH| < |TΔS|) for lower temperatures before becoming enthalpy-driven for higher temperatures. A negative ΔCp was obtained for all CIP-Ad/β-CD complexes (I to IV). The possible driving forces involved in the formation of the inclusion complex with CDs are numerous, such as van der Waals, hydrophobic, electrostatic, and hydrogen bonding, as well as charge–transfer interactions [39,58]. Moreover, other additional constraints, such as the release of conformational strain and the exclusion of cavity-bound high-energy water, can be implicated [39,58]. In the literature, a simple evaluation of enthalpy and entropy changes is usually used to distinguish the dominance of hydrophobic or van der Waals forces during complexation; hydrophobic interactions lead to ΔH > 0 and ΔS > 0 and van der Waals interactions lead to ΔH < 0 and ΔS < 0. In fact, interpretation of the thermodynamic parameters is more complex. One reason is that the formation of an inclusion complex is often accompanied by the release of water molecules from the CD cavity, which are at a higher level of energy in comparison with those in the external medium. This highly energetic exclusion phenomenon also contributes to the negative enthalpy and entropy changes observed [58]. Other authors traditionally admit that complexation is governed by hydrophobic interactions when the process is entropy-driven, with a larger and positive entropy and smaller enthalpy (|ΔH| < |TΔS| with ΔS > 0). By contrast, van der Waals interactions would usually be enthalpy-driven processes with minor favorable or unfavorable entropy interactions (|ΔH| > |TΔS| with ΔS > 0 or ΔS < 0). Consequently, hydrophobic and van der Waals interactions would be the two principal forces leading to the complexation for CIPAd/β-CDs (I to IV) complexes. However, a study on a large temperature range permitted the access of ΔCp and the discrimination of the major force. When a negative ΔCp is obtained, hydrophobic bonds are formed as a breakdown sequence of water clathrates around apolar molecules corroborating the desolvation hypothesis of β-CD [58]. In contrast, the hydrogen bond between β-CD and the guest makes a positive contribution to ΔCp [59]. Consequently, the large negative ΔCp values observed here for CIP-Ad/β-CD complexes confirmed the contribution of hydrophobic interactions as the main driving forces.

### 2.3. Solubility Studies

CIP-Ad **4** solubility was measured in water to evaluate its behavior near to physiological pH. Phase-solubility profiles are given in Figure 6A,B for native CD I and derived II, III, and IV, respectively. CIP-Ad is a poorly soluble drug with an intrinsic solubility (S_0_) in water determined around 1.10^−6^ M. The solubility of CIP-Ad significantly increased as a function of the CD concentration. According to Higuchi and Connors’s classification [60], the solubility curves can be considered as AL-type, that is, a linear increase in solubility of CIP-Ad versus CD concentration is obtained. When 1/1 complexation is supposed between the drug and CD, a linear correlation is attempted between the total solubility of the drug (S_t_) and the total concentration of cyclodextrin ([CD]_t_) in the aqueous medium (Equation (1)):(1)$$S_{t}\;=\;S_{0}+\frac{K_{1/1}{\;S}_{0}}{1+K_{1/1}{\;S}_{0}}{\left[\mathrm{CD}\right]}_{t}$$
where S_0_ is the intrinsic solubility of CIP-Ad (i.e., the solubility when non cyclodextrin is present) and K_1/1_ is the apparent stability constant of the drug/CD complex. The plot of St versus [CD]_t_ gives a straight line with a slope K_1/1_ S_0_/(1 +K_1/1_ S_0_)) less than unity and an intercept (S_int_) equal to S_0_. Next, K_1/1_ can be determined from the slope and S_0_ according to Equation (2):(2)$$K_{1/1}\;=\;\frac{\mathrm{Slope}}{S_{0}\left(1-\mathrm{Slope}\right)}$$

Equation (2) according to Loftsson et al. [61].

A linear fit with a slope less than unity was obtained for all CDs, in the range of CD concentrations studied, confirming the 1/1 stoichiometry of interaction. However, Sint values were far from S_0_ for all CDs and a negative value was obtained for host CD IV. This negative deviation from linearity at low CD concentration (i.e., S_int_ < S_0_), reported as the A-L-type phase-solubility profile in the literature, is currently observed for poorly soluble drugs (S_0_ < 1.10^−4^ M) [61]. It could be due to the nonideality of water as a solvent, the self-association of the drug molecules and of the drug/CD complexes, as well as non-inclusion complexation phenomena [61]. The use of incorrect S_int_ values results in an overestimation of K (or even negative K in the case of negative S_int_) [61]. Consequently, the determination of K was made here from an experimental value of S_0_. K values around 66.10^4^ M^−1^, 6.10^4^ M^−1^, 52.10^4^ M^−1^, and 37.10^4^ M^−1^ were obtained for I, II, III, and IV host molecules, respectively. In comparison with ITC results, slightly higher values (excepted for III) were obtained, which could be attributed to the more hydrophobic form of CIP-Ad **4** in water than in acidic buffer (that is, the net charge of the molecule is lower).

### 2.4. Antimicrobial Efficacy of Ciprofloxacin

In order to thereafter evaluate the antimicrobial efficacy of CIP against *S. epidermidis*, Minimal Inhibitory Concentration (MIC) values of the antibiotic were measured through the microdilution method based on ISO standard 20.776 [62]. Experiments were performed in microplates, filled with Tryptic Soy Broth (TSB), with an initial bacterial concentration of 10^6^ CFU/mL. Values ranging from 0.25 to 4 μg/mL were obtained for CIP depending on the tested strain (Table 3).

Strain 5 was the most sensitive and was consequently used for most of the further investigations. It is well known that biofilms correspond to high-cell-density systems. It is the reason why the efficacy of high CIP concentrations was tested against high-cell-density planktonic cultures. Experiments were performed in 96-well microplates containing TSB inoculated with an initial bacterial concentration of about 10^9^ CFU/mL. Bacteria were enumerated by plating out on Tryptic Soy Agar medium after incubation for 24 h at 37 °C. Time-killing curves are given in Figure 7. The exposure of planktonic cells to 10 × MIC (i.e., 2.5 μg/mL) of antibiotic during 24 h allowed a reduction of 3 log in the bacterial population.

The increase in the antibiotic concentration to 20 × MIC did not significantly improve the treatment efficacy. For this reason, the highest tested CIP concentration was set to 2.5 μg/mL for further experiments.

Biofilms were formed in 96-well microplates containing 200 μL of TSB, as previously described [63]. The ability of *S. epidermidis* species to produce β-(1-6)-*N*-acetyl-D-glucosamine polymer (PNAG) results from the presence of the *ica* locus, which controls the production of the polysaccharide intercellular adhesin (PIA) [1]. The detailed structural elucidation of PNAG has been so far performed on different *S. epidermidis* [64,65]. Strain 5 was isolated from an infected orthopedic prosthesis [66]. Biofilms formed by this strain contained considerable amounts of PNAG as testified by very similar 1H NMR data observed for the RP62A and 9142 strains [64,65].

The ability of strain 5 to form a biofilm was confirmed by measuring the sessile population in microplates after 24 h of incubation. The results demonstrated that the strain was able to form a biofilm, with the sessile population reaching about 1 × 10^9^ CFU/mL (SE n = 8). This bacterial concentration has been consequently used to test the different compounds on planktonic cells. The antimicrobial effect of CIP and DspB was then investigated on sessile microorganisms (strain 5). After 24 h of incubation, wells were washed with Phosphate-Buffered Saline and filled with 200 μL of TSB enriched with 5 μg/mL of DspB and/or increasing amounts of CIP. After 2 h of treatment at 37 °C, the number of adherent bacteria before and after treatment was compared. The results given in Figure 8 confirmed the high resistance of sessile organisms to CIP since only 1 log kill was observed after exposure for 24 h to a high antibiotic concentration (12.5 μg/mL, that is, 50 × MIC).

DspB used alone as a control did not exhibit an antimicrobial activity. DspB and CIP exerted a synergistic effect against the biofilm, confirming bibliographic data [28,29,30]. Used together at concentrations of 5 μg/mL of DspB and 2.5 μg/mL of CIP (that is, 10 × MIC), a bactericidal effect (5 log kill) was indeed observed on sessile organisms after exposure for 24 h. This result showed that DspB increases the bacterial susceptibility to CIP. Finally, MICs of CIP-Ad **4** were evaluated on the four *S. epidermidis* strains. The corresponding MICs are given in Table 4.

The results point out that, for the four strains used, an antibacterial activity of CIP-Ad **4** was observed, with higher MIC values than those measured for CIP, however (Table 4). Here, again, strain 5 was the most sensitive strain.

### 2.5. Combined Effect of Dsp B and β-CD II/CIP-Ad Complex against Biofilms

The limited bibliographical data pertaining to the CIP interaction with CDs (e.g., dissolution, absorption) makes the prediction of the drug delivery processes difficult. Several mechanisms can contribute to drug release after parenteral administration [67,68], such as drug dilution, competitive displacement by endogenous materials, drug binding to plasma and tissue components, and drug uptake into nontargeted tissues. The pharmacokinetic drug properties should be unaffected by the use of CDs, when the binding constant is below 10^5^ M^−1^ [69]. Consequently, two candidates appeared particularly suitable for further studies: the 2,3-*O*-dimethyl-β-CD (II)/CIP-Ad and the 2,3,6-*O*-trimethyl-β-CD (IV)/CIP-Ad complexes.

MICs of the two complexes against the four strains are given in Table 5. The results showed that the complexation of CIP-Ad **4** with the 2,3-*O*-dimethyl-β-CD (II) did not alter the CIP-Ad derivative activity, while complexation with the permethylated β-CD (IV) increased the MIC value on the four tested strains. For these reasons, the 2,3-*O*-dimethyl-β-CD (II)/CIP-Ad complex was used for further experiments. Again, strain 5 exhibited a higher susceptibility.

The susceptibility of planktonic bacteria against the β-CD II/CIP-Ad complex was then evaluated as described above. The concentrations used corresponded to 2.5× and 10 × MIC CIP-Ad. The results (Figure 9) showed an expected increase in the antimicrobial activity as the complex concentration increased, with the 24 h of exposure to the complex at 10 × MIC leading to more than a 3log unit reduction in the bacterial population.

Before investigating the antimicrobial activity of the DspB/β-CD II/CIP-Ad complex association, on sessile organisms, the efficacy of the complex alone was evaluated (Figure 10).

The results showed that this complex exhibited the same activity against sessile organisms than the CIP (see Figure 8). For example, about a 1 log unit decrease in the bacterial population was observed for 50 × MIC. When used in association with 5 μg/mL of DspB, a reduction of 2 log unit in the sessile population was observed after 24 h of exposure (Figure 10). These data confirmed the positive effect of the association, with a control experiment confirming the absence of antimicrobial activity of DspB when used alone.

### 2.6. Synthesis of CD Derivatives CD1-3

In order to elaborate DspB-β-CD conjugates, we synthesized three carboxylic CD derivatives with different spacer arm lengths, i.e., with 2, 4, and 10 atoms of carbon, named CD1, CD2, and CD3, respectively (Figure 2). We focused this preliminary study on permethylated cyclodextrins, because they are easier to purify in large quantity. From this 6-*O*-monofunctionalization, the DspB was grafted via an amidation reaction (Figure 3). The strategy used for the functionalization of the β-CD core involved the synthesis of the well-known mono-6-*O*-tosyl-β-CD precursor **5** [70]. After the introduction of the azide function, the macrocycle **6** was permethylated as described in the literature. A reduction step afforded the monoamino derivative **8** [70], ready for a coupling reaction with succinic anhydride [71], monomethyl adipic acid ester **a**, or monomethyl dodecanedioic acid ester **b** at room temperature [45]. The corresponding 6-monoalkylcarboxyl-β-CD derivatives, **CD1**, **CD2**, and **CD3**, were obtained after a saponification step.

The immobilization of the DispersinB^®^ on the three macrocycles (CD1, CD2, and CD3) was performed with 1-ethyl-3-(3-dimethylaminopropyl)carbodiimide hydrochloride (EDAC) and *N*hydroxysuccinimide (NHS) in borate buffer (Figure 3). After ultrafiltration of the enzymatic solution, the modified DspB was purified using a 2-D Clean-Up Kit (GE Healthcare).

Mass spectrometry analyses confirmed the expected masses of 69,994, 48,106, and 51,171 g/mol for **CD1**, **CD2**, and **CD3** with maximum rates of 17, 4, and 9 molecules per enzyme, respectively. Native DspB was never found on MS spectra, suggesting a high grafting rate of DspB onto modified CDs.

### 2.7. Enzymatic Biofilm Dispersion Assays by DspB-β-CD Conjugates

Tests were adapted from the method implemented by O’Toole and Kolter [72]. Briefly, bacteria were grown for 24 h in TSB at 37 °C in microplates. Unattached cells were removed by rinsing the wells thoroughly with PBS, and biofilms were subsequently stained by incubation with Crystal Violet (CV). CV was then solubilized by adding ethanol and the OD of the solution measured at 550 nm.

The antibiofilm activity of the three DspB-β-CD conjugates (that is, bearing the three different arms) is given in Figure 11. The results showed that the conjugate bearing the shorter arm (CD1) exhibited a lower activity (highlighted by high OD values), whereas no alteration of the enzyme activity was observed with the two other conjugates (CD2 and CD3), as compared with DspB alone. As expected, this activity was concentration-dependent. The enzymatic alteration observed with CD1 is probably due to a steric effect, the high number of CDs grafted onto the enzyme (17 per enzyme molecule), and the proximity of CDs restricting substrate accessibility to the active sites.

These results were then confirmed on the three other *S. epidermidis* strains (Figure 12). The results showed that the conjugate was efficient on all biofilms, confirming that the three strains produced PNAG. Here, again, the DspB-CD1 conjugate exhibited the lowest antibiofilm activity.

### 2.8. Effect of the DspB-β-CD3/CIP-Ad Conjugate on Biofilm of S. epidermidis

Finally, we tested the antimicrobial activity of the drug delivery system DspB-β-CD3/CIP-Ad conjugate on biofilms of strain 5. A CIP concentration of 5 × MIC, i.e., 10 μg/mL, was used. A significant activity of the drug delivery system was observed (Figure 13) on a 24 h-old biofilm, since about a 3-log reduction in the sessile bacterial population was observed after 24 h of exposure. This activity was logically lower than that recorded with the association of DspB/CIP (Figure 8) as a lower CIP concentration was used. The efficiency of the drug delivery system was similar to that obtained when the 2,3-*O*-dimethyl-β-CD (II)/CIP-Ad complex was used in association with the DspB (Figure 10), which demonstrates that the chemical immobilization of CDs on DspB does not alter its antibiofilm properties.

## 3. Materials and Methods

Solvents (HPLC quality) and reagents β-CD, ciprofloxacin and ciprofloxacin hydrochlorides were purchased from Sigma Aldrich (France, St. Louis, MO, USA). Deuterated solvents were purchased from Euriso-top. Thin layer chromatography (TLC) was performed on (silica gel 60 F254 aluminium-backed plates) and (silica gel 60 RP-18. F254s) purchased from Merck KGaA, Germany Spot detection was carried out with (ethanol/H_2_SO_4_) 92:8, nihydrine, UV light (λ = 254 nm), Fluorescence light at 365 nm, Typical column chromatography on silica gel was used and chromatography on silica gel RP-18 using strata C18-E colon purchased from phenomenex. Electrospray mass spectrometry (ESI) spectra were recorded on an Esquire-LC ion-trap mass spectrometer (ITMS) equipped with an ESI source and the Esquire control 6.16 data system. The 1H NMR spectra were measured on a Bruker AC 300 spectrometer. Infrared analysis was done on spectrometer 100 FT-IR (Perkin-Elmer). Elementary analysis was realized by the laboratory of micro-analyses (IRCOF, University of Rouen, France). High–resolution mass spectra were performed on a LC-TOF Premier XE (Micromass, Manchester, UK) equipped with ESI source.

### 3.1. Synthesis of β-CD Derivatives

β-Cyclodextrin was dried overnight at 100 °C in vacuum prior to use. Pyridine and N,N-dimethylformamide (DMF) were freshly distilled from CaH_2_. All other reagents were of the highest available commercial quality and used without further purification.

Heptakis (2,3-O-dimethyl)-β-cyclodextrin II Compound **3** was prepared as described by [38].

Heptakis (2,6-O-dimethyl)-β-cyclodextrin III Compound **3** was prepared as described by [39].

Permethyl mono-6-amino-β-cyclodextrin IV Compound **4** was prepared as described by [38].

### 3.2. Adipic Acid Monomethyl Ester (a)

Dimethyl adipate was obtained by reaction of adipic acid, (10 g, 43.45 mmol) in MeOH (34.09 mL) was added 0.75 mL of H_2_SO_4_ the mixture was refluxed overnight. The solution was concentrated in vacuo. The resulting precipitate was poured on ice, and extracted with Et_2_O (3 × 40 mL) yield (11.2 g, 100%). KOH was added to the obtained compound in MeOH. Extensive washing with hexane removed the dodecanedioic acid monomethyl ester. Evaporation of the washing gave the title compound (b) as oil with 60% yield (4.1 g).

Dodecanedioic acid Monomethyl Ester (b): The compound b was prepared from dodecanedioic acid following the same procedure described for compound a.

Permethyled β-cyclodextrin derivative CD1 Compound **5** was prepared as described by [40].

### 3.3. Permethyl β-Cyclodextrin Derivative CD2

The compound 4 (0.33 g, 0.23 mmol) were dissolved in 40 mL of anhydrous DMF, and then reacted with DIC (519.7 μL, 3.30 mmol) and HOBt (0.5 g, 3.58 mmol) for 2 h at room temperature. A solution of adipic acid monomethyl ester (0.04 g, 0.27 mmoL) was added in 10 mL of dichloromethane with a few amounts of triethylamine The mixture was stirred for 24 h at room temperature. The solvent was removed under vacuum. The residue was extracted in dichloromethane (20 mL) and washed with water (2 × 5 mL). The combined organic phase was dried in MgSO_4_ and evaporated under vacuum. The crude products were purified by chromatography on silica gel column eluting with DCM:MeOH, 7:3. The pure fractions were combined and then concentrated in vacuo to give the desired compound with 71% yield (0.26 mg).

To a solution of obtained compound (0.2 g, 0.2 mmol) in 0.4 mL of MeOH, was added 0.1 mL of aqueous solution of KOH (1 M) and then the reaction mixture was stirred at room temperature for 1 h. The TLC showed the completely desperation of the reagent. The reaction was stopped by removing of MeOH under vacuum. Then the residue was dissolved in 20 mL of water. The solution was then acidified by adding few mL of hydrochloric acid (1 M) at pH = 2 to obtain the acidic product form. The product was extracted with (3 × 40 mL) dichloromethane; the combined organic phase was evaporated under vacuum The residue was taken up in ethanol (5 mL), filtered and the filtrate was then evaporated under vacuum and dried in vacuum at 80 °C for 24 h. The desired product CD2 was obtained as white solid (0.2 g, over yield 71%).

TLC: Rf 0.052 (DCM/MeOH:95/5); mp (°C): 138; RMN 1H (300 MHz, CDCl3): δ (ppm) 1.60–1.61 (m, 4H, (-CH2)2-CH2-COOH), 2.15–2.25 (m, 4H, -CH2-COOH), 3.13 (m, 1H, -CH-CH), 3.31 (s, 3H, OCH3), 3.34 (m, 1H, CH-CD), 3.43 (s, 3H, OCH3), 3.54 (m, 1H, CH-CD), 3.56 (s, 3H, OCH3), 3.73–3.75 (m, 2H, CH2-OCH3); RMN 13C (75 MHz, CDCl3): δ (ppm) 24.5–25.2 (-CH2-(CH2)2-COOH), 33.3 (-CH2-COOH), 36.1 (-NH-CO-CH2-COOH), 39.9 (-CH2-NH-CO-), 58.5 (-OCH3), 58.9 (-CH2-OCH3), 61.8 (-CH-OCH3), 70.8 (-CH-O-CH), 71.3 (-CH2-OCH3), 80.3 (-CH-CH-OCH3), 81.5 (-CH-CH-OCH3), 81.7 (-CH-CH-OCH3), 98.5–98.9 (-CH-O-); ESI-MS: calcd for C68H119NO37 1540,75. Found 1540.73 [M-H]-; HR-MS: TOF-MS-ES- *m*/*z* = 1540.73 [M-H]-, M = 1541.75 g/mol^−1^.; IR ν(cm^−1^): ring vibration 946 cm^−1^, (C-C/C-O) 1032–1079,(C-H/O-H) ν = 1297–1449,(CO-NH) 1731,(CO-OH) 1774, (CH2) 2931, (N-H) 3405. 

### 3.4. Permethyl β-Cyclodextrin Derivative CD3 

This compound was obtained as the same procedure described above. The compound 4 (1.2 g, 0.8 mmol) was added to the compound b (0.05 g, 51.5 mmol). The desired product CD3 was obtained as white solid (0.05 g, over yield 84%).

TLC: Rf = 0.064 DCM/MeOH (95:5); RMN 1H (300 MHz, CDCl3): δ (ppm) 1.05–1.09 (m, 4H, -(CH2)3-(CH2)3-COOH), 1.21–2.25 (m, 4H, -(CH2)2-(CH2)2-COOH), 1.55 (m, 2H, -CH2-CH2-COOH), 2.12–2.23 (m, 4H, -CH2-COOH), 3.13 (m, 1H, -CH-OCH3), 3.32 (s, 3H, -CH-OCH3), 3.36 (m, 1H, -CH-OCH3), 3.43–3.44 (3H, s, -CH2-OCH3), 3.46 (m, 1H, -CH-OCH3), 3.54 (m, 1H, -CH-OCH3), 3.57–3.58 (s, 3H, -OCH3), 3.70–3.75 (s, 3H, -CH2-OCH3), 5.04–5.10 (m, 1H, -CH-O-CH-); RMN 13C (75 MHz, CDCl3): δ (ppm) 23.2–25.8 (-(CH2)3-(CH2)3-COOH), 24.8 -(CH2-CH2-COOH), 29.1–29.5 (-(CH2)2-(CH2)2-COOH), 34.2 (-CH2-COOH), 36.7 -(CH2-CO-NH), 39.9 (-CONH-CH2-OCH3), 58.3–58.6 (-OCH3), 58.8-59.9 (-OCH3), 61.2–61.5 (-OCH3), 71.0 (-CH-OCH3), 71.2–71.3 (-CH2-OCH3), 79.9–80.6 (-CH-OCH3), 81.3–82.2 (-CH-CH-OCH3), 98.8–99.0 (-CH-O-CH-), 173.5 (-CONH-), 177.4 (-COOH); ESI-MS calcd *m*/*z* = 1625, 84 [M-H]-, mass found *m*/*z* = 1625.0 [M-H]-; IR ν(cm^−1^): ring vibration 946 cm^−1^, (C-C/C-O) 1032–1079, (C-H/O-H) 1243–1436, (CO-NH) 1574, (-CO-OH) 1628; (CH2) 2849–2928, (N-H) 3329; HR-MS: TOF-MS-ES- *m*/*z* = 1625.83 [M-H]-, M = 1626.83 g/mol^−1^.

### 3.5. Synthesis Cirpofloxacin-Adamantyl Derivative

N-Boc protection of ciprofloxacin 2.

The mixture containing ciprofloxacin (3.61 g, 8.42 mmol), Boc2O (2.03g, 9.3 mmol) and 16.9 mL du NaOH aqueous solution in THF was stirred at room temperature overnight. The product precipitated from the reaction mixture and was collected by filtration as white solid, 100% yield (3.5 g).

TLC: Rf 0.55, DCM/MeH 90:10; 1H NMR (300 MHz, CDCl3): δ 1.19–1.20 (m, 2H), 1.36–1.43 (m, 2H), 1.49 (s, 9H), 3.27–3.30 (t, J = 7.13, 13.14 Hz, 4H), 3.65–3.68 (t, J = 4.8, 10.2 Hz, 4H), 7.34–7.36 (d, J = 7.11 Hz, 1H), 7.97–8.20 (d, J = 13.4 Hz, 1H), 8.74 (s, 1H), 14.94 (s, 1H). 13C NMR (75 MHz, CDCl3): δ 8.2 (CH2-cyclopropyl), 28.4 (C-Boc), 35.3 (CH-cyclopropyl), 49.9 (CH2-piperazin), 104.7 (CH-Ar), 107.9 (C-Ar), 111.7 (CH-Ar), 119.9 (C-Ar), 138.8 (C-Ar), 146.4 (C-Ar), 152.1 (C-Ar), 155.2 (C-F), 167.2 (-COOH), 177.3 (Ar-C=O), 177.3 (Ar-C=O), 177.3 (-N-C=O); ESI-MS: calcd for (C28H33FN4O4+Na)+: 431.19. Found: 454.0 [M+Na]+.

7-(4-((1S, 3R, 5S))-adamantane-1-carbonyl) piperazin-1-yl)-1-cyclopropyl-6-fluoro-4-oxo-1,4 dihydroquinoline-3-carboxylic (tert-butyl carbonic) anhydride 2

To a solution of the compound 2 (1.4 g, 3.3 mmol) in DMF (16.7 mL), HBTU (2.5 g, 6.6 mmol) was added followed by HOBt (0.4 g, 3.3 mmol), DIPEA (0.6 g, 5 mmol), and 1-amino adamantane (0.604 g, 4 mmol). The mixture was stirred at room temperature under N2 for 30 min, and further DIPEA (0.85 mL, 5 mmol) was added. The reaction mixture was stirred at room temperature for 1 h, poured into water, and extracted with CH2Cl2. The organic layer was washed with brine, dried over anhydrous Na2SO4, and evaporated. The crude residue was purified by column chromatography on silica gel using DCM:MeOH 5:95 as eluant to provide the compound 3 (90% yield) as yellow solid.

TLC: Rf 0.34 (DCM/MeOH 95:5); mp (°C): 232; 1H NMR (300 MHz, CDCl3): δ (ppm) 1.14–1.16 (m, 2H), 1.28–1.31 (m, 2H), 1.48 (CH3, s, 9H), 1.65–1.71 (CH2-Ad, CH-Ad, m, 7.88), 2.09 (CH-Ad, 3H), 2.15 (CH2-Ad, m, 4.88 H), 3.20–3.24 (CH2-piperazin, m, 2H), 3.40–3.46 (CH-pentyl, m, 1H), 3.63–3.66 (CH2-piperazin, m, 2H), 7.29–7.32 (H-Ar, d, J = 7.11 Hz, 1H), 8.01–8.05 (H-Ar, d, J = 13.23 Hz, 1H), 8.80 (H-Ar, s, 1H), 9.86 (NH, s, 1H); 13C NMR (75 MHz, CDCl3): δ (ppm) 8.0 (CH2-cyclopropyl), 28.4 ((CH3)3-C-BOC), 29.5 (Ad-CH), 34.6 (cyclopropyl-CH), 36.5 (Ad-CH2), 41.7 (Ad-CH2), 42.5 (CH2-piperazin), 49.9 (CH2-piperazin), 104.8 (CH-Ar), 112.3 (CH-Ar), 112.6 (CH-Ar), 146.4 (C-Ar). 155.2 (C-F), 154 (Boc-C=O), 163.6 (NH-C=O), 175.7 (Ar-C=O); ESI: calcd calcd for C32H41FN4O4 *m*/*z* = 564.31 found: 587 [M+Na]+; 19F NMR (282 MHZ, CDCl3): δ (ppm) −123.33 (F-Ar), −113.36 (F-Ar); IR ν (cm^−1^): (C=C), 1656, (C=O) 1698, (CH2) 2231, 2906, (CH2), (N-H) 3439; Anal. Calcd/Found: calcd for (C27H33FN4O2): C, 69.80; H, 7.16; N, 12.06; found 62.80, 6.65, 9.77; 

HR-MS: TOF-MS-ES- *m*/*z* = 565.31 [M+H]+, M = 565.31 g/mol^−1^.

7-(4-((1S, 3R, 5S)-adamantane-1-carbonyl) piperazin-1-yl)-1-cyclopropyl-6-fluoro-4-oxo-1,4 dihydroquinoline-3-carboxylic acid 4.

Cleavage of Boc group was achieved by treatment of conjugate 3 (4.3 g, 0.7 mmol) with 10% TFA solution in dichloromethane (19.7 mL). Complete reaction was confirmed by TLC. The compound was extracted with 25 mL of AcOEt and organic phase was then collected and evaporated under vacuum. The resulting residue was washed three times with 200 mL of hexane, dissolved in minimum volume of methanol, precipitate by addition of cold diethyl ether and kept at −20 °C overnight. The solid precipitate was obtained by decantation after solvent removing by high vacuum. The residue was recovered in 200 mL of water and the pH was then adjusted to 9 by dropwise adding of aqueous 1 M NaOH solution. The desired product was precipitated and extracted by DCM to eliminate the trifluoroacetic acid sodium salt. The organic layer was washed with brine, dried over anhydrous Na2SO4 and evaporated to dryness.

TLC: Rf 0.27 (DCM/MeOH 95:5); mp (°C): 232; 1H NMR (300 MHz, CDCl3): δ 1.1–1.14 (Bd, 2H), 1.24–1.28 (Bd, 2H), 1.67 (CH2, s, 6H), 2.06 (CH, s, 3H), 2.11 (CH2, s, 6H), 3.06–3.09 (CH2, m, 4H), 3.22–3.26 (CH2, m, 4H), 3.40–3.45 (CH, m, 1H), 7.26–7.28 (CH, d, J = 7.5 Hz, 1H), 7.92–7.96 (CH, d, J = 13.15 Hz, 1H), 8.74 (CH, s, 1H), 9.86 (NH, s, 1H); 13C NMR (300 MHz, CDCl3): δ 8.0 (CH2), 28.5 (Ad-CH), 33.7 (cyclopropyl-CH), 35.5 (Ad-CH2), 40.7 (Ad-CH2), 44.7.

(piperazin-CH2), 49.8 (piperazin-CH2), 103.58–103.62 (CH-Ar), 112.2 (CH-Ar), 112.5 (CH-Ar), 121.8 (C-Ar), 138.2 (C-Ar), 145.1 (CH-Ar), 151.4 (C-Ar), 155.2 (C-F), 163.6 (NH-C=O), 175.7 (Ar-C=O), 19F (400MHZ, CDCL3): δ 123.36 (s, 1F); ESI-MS: calcd for (C27H33FN4O2+H)+: 465.30; found: 465.40 [M+H]+; HR-MS: TOF-MS-ES- *m*/*z*= 464.26 [M+H]+, M = 465.26 g/mol^−1^; IR ν (cm^−1^): (C=C), 1625, (C=O) 1297–1449, (CH2) 2349, 2844, 2908, (CH2) 2931, (N-H) 3446.; Anal. Calcd/Found: calcd for (C27H33FN4O2): C, 69.80; H, 7.16; N, 12.06; found 62.80, 6.65, 9.77.

### 3.6. Isothermal Titration Calorimetry (ITC)

Calorimetric studies were performed using the Nano ITC 2G Isothermal Titration Microcalorimeter from TA instruments (New Castle, DE, USA). This power compensation differential instrument has a reference cell and a sample cell of approximately 1.040 mL. Each experiment involved 25 injections of 10 μL of β-CD or derivatives into the thermostated sample cell containing solution of CIP-Ad. The reference cell was filled with degassed MilliQ-water. The time interval between two consecutive injections was fixed to 300 s and the agitation speed (via the paddle stirrer at the end of the syringe) at 250 rpm. Solutions of βCD or derivatives (4mM) and CIP-Ad (0.3 mM) were prepared in chloroacetate buffer (pH = 3, 0.05 M) and filtered through 0.2 μm cellulose acetate membrane (Millipor^®^, France) before use. Experiments were realized at atmospheric pressure and at 310 K. For all experiments, corrections have be made to the heat data to account for heat effects associated with titrant dilution and any temperature difference between titrant and titrate solutions. These corrections were made by performing a blank titration experiment and subtracting the blank heat data from the experimental thermogram. The blank titration was carried out by introducing βCD or derivatives (4 mM) in the sample cell containing buffer alone (without CIP-Ad), according the same experimental conditions (volume, agitation speed, temperature). All experiments were reproduced at least two times. The binding isotherms (without the first data point) were analyzed using an independent binding model of NanoAnalyse software (which models an interaction of “n” ligands with a macromolecule that has one binding site) to estimate the thermodynamic parameters of interaction such as the apparent binding stoichiometry (n), association constant (Ka) and enthalpy of binding (ΔH). Then, since temperature is held constant throughout, the free Gibbs energy (ΔG) of the binding reaction was determined by ΔG = −RTlnKa. Consequently, the entropy change (ΔS) was determined by ΔS = (ΔH − ΔG)/T.

### 3.7. NMR Study

Preparation of samples for NMR analysis.

The CIP-Ad sample was prepared by dissolving an excess amount of CIP-Ad in D2O then the solution was sonicated using powerful ultrasonic bath for 2 h. The suspension was filtrated through a 0.2 μm sy-ringe filter (Millipor®, France) to eliminate the insol-uble supernatant. The filtrate was analyzed by NMR. A solution of 2,3-O-dimethyl-β-CD at 4 mM in D2O was prepared.

Preparation of Complex between 2,3-di-O-methyl-β-CD and (CIP–Ad).

2 equivalents of 2,3-O-dimethyl-β-CD (10.64 mg, 8 mM) were added to 1 equivalent of CIP–Ad (1.85 mg, 3.98 mmol) in 1 mL of D2O. The cloudy solution obtained was stirred and then filtrated through a 0.2 μm syringe filter (Millipor®, France) to eliminate the insoluble supernatant. The filtrate was analyzed by NMR. The reactants, 2,3-O-dimethyl-β-CD (in D2O) and CIP-Ad (in D2O and CDCl3), were analyzed with a Bruker AVIII300 NMR spectrometer. Experiments were carried out at 300 MHz for 1H and at 75 MHz for 13C. The complete assignment of all H and C atoms of 2,3-O-dimethyl-β-CD and CIP-Ad was achieved by conventional 2D experiments: HMQC (1H-13C) and HMBC (1H-13C). For 2D spectra, a total of 1024 (HMQC) or 2048 (HMBC) points in F2 and 128 experiments in F1 were recorded. The complex 2,3-O-dimethyl-β-CD:CIP–Ad (in D2O) was analyzed with a Bruker AVIII600 NMR spectrometer equipped with a 10 A gradient amplifier and a 5 mm CPTXI (1H, 13C, 15N) including shielded z-gradients. 1D 1H and 2D 1H-1H NMR were recorded at 298 K. All chemical shifts were measured relative to external TMS. 2D NOESY (1H-1H): was used to evidence the interaction between 2,3-O-dimethyl-β-CD and CIP–Ad. For 2D NOESY (1H-1H) spectrum, a total of 2048 points in F2 and 256 experiments in F1 were recorded. A mixing time of 300ms was used. The formation of the complex was confirmed using the diffusion coefficients measured by 1H DOSY experiments. The diffusion 1H NMR experiments were carried out by using the AVIII600 spectrometer, for 2,3-O-dimethyl-β-CD, CIP-Ad and complex. A pulsed-gradient stimulated echo (LED-PFGSTE) sequence, using bipolar gradient was used. Sequence delays were: diffusion delay (Δ) 150 ms, recovery delay after gradient (τ) 0.2 ms, and LED recovery delay 5 ms. For each data set, 66,560 complex points were collected for each 32 experiments in which the gradient strength was linearly incremented from 0.681 to 32.35 G.cm^−1^. The gradient duration δ/2 was adjusted to observe a near complete signal loss at 32.35 G.cm^−1^. The δ/2 delay was fixed at 2 ms. A 2s recycle delay was used between scans for all data shown. The number of scans was fixed to 32 (for the free molecules) and 64 scans (for the complex). The number of scans was fixed to 32 (for the free molecules) and 64 scans (for the complex).

TLC: Rf = 0.064 DCM/MeOH (95:5); RMN 1H (300 MHz, CDCl3): δ (ppm) 1.05–1.09 (m, 4H, -(CH2)3-(CH2)3-COOH), 1.21–2.25 (m, 4H, -(CH2)2-(CH2)2-COOH), 1.55 (m, 2H, -CH2-CH2-COOH), 2.12–2.23 (m, 4H, -CH2-COOH), 3.13 (m, 1H, -CH-OCH3), 3.32 (s, 3H, -CH-OCH3), 3.36 (m, 1H, -CH-OCH3), 3.43–3.44 (3H, s, -CH2-OCH3), 3.46 (m, 1H, -CH-OCH3), 3.54 (m, 1H, -CH-OCH3), 3.57–3.58 (s, 3H, -OCH3), 3.70–3.75 (s, 3H, -CH2-OCH3), 5.04–5.10 (m, 1H, -CH-O-CH-); RMN 13C (75 MHz, CDCl3): δ (ppm) 23.2–25.8 (-(CH2)3-(CH2)3-COOH), 24.8 -(CH2-CH2-COOH), 29.1–29.5 (-(CH2)2-(CH2)2-COOH), 34.2 (-CH2-COOH), 36.7 -(CH2-CO-NH), 39.9 (-CONH-CH2-OCH3), 58.3–58.6 (-OCH3), 58.8–59.9 (-OCH3), 61.2–61.5 (-OCH3), 71.0 (-CH-OCH3), 71.2–71.3 (-CH2-OCH3), 79.9–80.6 (-CH-OCH3), 81.3–82.2 (-CH-CH-OCH3), 98.8–99.0 (-CH-O-CH-), 173.5 (-CONH-), 177.4 (-COOH); ESI-MS calcd *m*/*z* = 1625, 84 [M-H]-, mass found *m*/*z* = 1625.0 [M-H]-; IR ν(cm^−1^): ring vibration 946 cm^−1^, (C-C/C-O) 1032–1079, (C-H/O-H) 1243–1436, (CO-NH) 1574, (-CO-OH) 1628; (CH2) 2849–2928, (N-H) 3329; HR-MS: TOF-MS-ES- *m*/*z*= 1625.83 [M-H]-, M = 1626.83 g/mol^−1^.

### 3.8. Test of CIP-Ad Solubility

Triplicate saturated solutions of CIP-Ad in water were prepared by adding excess amount of compound in distilled water. Flasks with concentrated solutions were capped with plastic stoppers and mechanically shaken for 24 h at 25 °C and 250 rpm. The solutions were then taken up into syringe filter, and filtered through 0.2 μm microcellulose filter (Merck Millipore, Guyancourt, France). UV measurements (optical density, 280 nm) were performed to quantify the CIP-Ad amount in solutions (Perkin Elmer Carry 100 BIO UV-vis spectrophotometer) by using a calibration curve (R^2^ = 0.99).

Phase solubility test: The solubility tests were performed with CIP-Ad in presence of three methylated β-CD derivatives at various concentrations (0; 3; 12; 40; 75; 110; 155; 250 mM) using 5 mL glass flasks. 13.96–14 mg of CIP Ad was added to the aqueous CDs solution for a concentration of 14 mg/L. The tubes were recapped and mechanically shaken for 48 h at 37 °C and 250 rpm. After 48 h, the solution was extracted using a 1 mL syringe and filtered with a 0.2 μm microcellulose filter. 0.5 ML of sample was diluted with 0.5 mL of ethanol in order to dissociate the μm microcellulose filter. 0.5 ML of sample was diluted with 0.5 mL of ethanol in order to dissociate the complex. Dilution (1/500) of each sample with 50% aqueous ethanol was necessary to performed UV absorption spectrophotometry and quantified the CIP-Ad in solution. A 50% aqueous ethanol solution was used as a blank. For solubility test with β-CD, the same procedure was followed. β-CD was added to four tubes at various concentrations in a range of 2; 4; 6; 8; 12; 16 mM. The experiment was performed in triplicate (n = 3).

Preparation of Solid Inclusion Complex: The inclusion complex was prepared by freeze-drying method water by adding CIP-Ad into CD solution prepared using Milli-Q water. The concentrations of each component were calculated at the highest solubility given by the solubility curves. The aqueous CIP-Ad mixture was shaken for 1 night at −4 °C at 250 rpm (Innova 4230, New Brunswick Scientific, Edison, NJ, USA). The solution was filtered and placed into wide shallow dishes covered with plastic wrap and frozen to −80 °C. Samples were freeze-dried over four days (Sentry Freezemobile 12SL, Virtis, Gardiner, NY, USA). The solid sample was stored in sealed bottles at −4 °C until analysis.

Preparation of Dispersin B–CD conjugates: EDAC (30 mg) and 16 mg of N-hydroxysuccinimide were added to the reaction mixtures containing 0.5 mg of Dispersin B dissolved in 0.5 mL of 80 mM borate buffer, pH 8.0, and 76 mg of each CD derivative, i.e, CD1, CD2, and CD3. The solutions were stirred for 1 h at room temperature and then for 16 h at 4 °C. Ultrafiltration of enzymatic solution was performed using centrifugal filter (Amicon Ultra-15 Centrifugal Filter Unit with Ultracel-10 membrane, Millipore Merck, Guyancourt, France) for concentrating and desalting the enzyme and elimination of the entire reagent using a cut off 10 KDa. The modified Dsp B was purified by coprecipitation using a 2-D Clean-Up Kit (Biorad, Marnes-la-Coquette, France).

Minimal Inhibitory Concentration (MIC) determination: Antimicrobial activities of compounds were evaluated by MIC measurements through the micro dilution method based on ISO standard 20776-1 [41]. 

Briefly, microplate wells containing TSB were enriched by a solution of CIP hydrochloride at various concentrations between 1024 and 0.25 μg/mL in 0.15 M, Ph = 7.4 phosphate buffer (PBS). Each well was inoculated by a bacterial suspension (final concentration, 106 colony forming units CFU/mL). After 24 h of incubation at 37 °C, the bacterial growth was visualized by the broth turbidity. MIC was defined as the lowest concentration of CIP-Ad that inhibited the visible bacterial growth. Negative (without bacteria) and positive controls (without antibiotic) were included in each assay. The same procedure was used for CIP-Ad and each CDs complexes, i.e., CIP-Ad:2,3-O-dimethyl-β-CDs and CIP-Ad:permethyl β-CDs. In these cases CIP-Ad was solubilized in steril dimethylsulfoxide DMSO and CDs complexes in PBS.

Biofilm formation assay: Strain 5 was used. Biofilms were formed in 96-well microplates containing TSB inoculated from a preculture (initial cell concentration, 107 CFU/mL) as previously described [42]. After 24 h of incubation, the medium was removed and cells washed three times with PBS. Biofilm cells were then recovered by scrapping, re-suspended in sterile water and enumerated by plating out on Tryptic Soy agar medium.

### 3.9. Enzymatic Biofilm Dispersion Assays

Tests were adapted from the method described by O’Toole and Kolter [43]. Biofilms were formed as described above. After 24 h of incubation, unattached cells were removed by rinsing the microdishes thoroughly three times with PBS. Two hundred μl of solutions of Dsp B (final concentration, 5 μg/mL) or DspB-CD in PBS were added. After 2 h of incubation at 37 °C, biofilms were washed again and attached cells were subsequently stained by incubation with 0.1% Crystal violet (CV, Sigma-Aldrich, Singapore) for 30 min. The CV was then solubilized by adding 200 μL of ethanol/aceton (80/20, *v*/*v*) and the OD measured at 590 nm (Victor3 microplate reader, PerkinElmer Sciex). The OD590 values obtained at times zero (before DspB exposure) and after treatment were compared. For each strain the experiment was performed 3 times.

Antibiotic resistance assays: Biofilms were formed in 96-well microplates as described above. After 24 h of incubation at 37 °C, the medium was removed and cells washed three times with PBS. Wells were then filled with 200 μL of TSB containing different concentrations of ciprofloxacin, i.e., 5 MIC.

Biological tests.

Bacterial strains: *S. epidermidis* clinical strains (1457, 9142, RP62A and 5) were kindly provided by Pr. J. Kaplan from the University of Medicine and Dentistry of New Jersey (USA). Strains were stored at −80 °C in glycerol and used as required. For precultures, bacteria were grown for 24 h in Tryptic soy broth (TSB) (Difco, Beckton Dickinson, Franklin Lakes, NJ, USA) at 37 °C with shaking. All antibiotics, CIP or CIP-Ad was conserved at −4 °C.

Bacterial growth monitoring: The growth curve was obtained by incubating the strain 5 in TSB for 26 h at 37 °C and monitoring the OD595 (Figure 12). 10 MIC, 20 MIC, 50 MIC and 100 MIC. After two hours of incubation at 37 °C, the culture medium was removed and the number of adherent cells was enumerated as described above. The number of adherent cells before and after treatment was compared. A control without antibiotic was performed. For each strain, the experiment was performed 3 times.

The same protocol was used for the complex CIP-Ad:2,3-O-dimethyl-β-CD by using concentrations corresponding to 5 MIC, 10 MIC, 20 MIC, 50 MIC. The solid CIP-Ad:2,3-O-dimethyl-β-CD complex was prepared as described above and solubilized in PBS.

Combined effect of Dsp B and ciprofloxacin on established biofilms of *S. epidermidis*:

Biofilms were performed as described above. After 24 h of incubation at 37 °C, the supernatant was gently aspirated with a micropipette in each well. Wells were then washed three times with PBS under aseptic conditions to eliminate the unbound bacteria, without disturbing the adherent film. 

Ciprofloxacin (10 × MIC) and/or, Dsp B (5 μg/mL) were added in 200 μL PBS. Plates were incubated at 37 °C under shaking. After 2 h of incubation, the growth medium was removed and wells washed three times with PBS. Biofilm cells were recovered by scrapping and enumerated as described above.

Combined effect of DspB with (CIP-Ad:2,3-O-dimethyl-β-CD) complex, Dsp B-CD3 conjugate and Dsp B-CD3:CIP-Ad conjugate on biofilms.

The same procedure was used to test the effect of the Dsp B CIP-Ad:2,3-O-dimethyl-β-CD complex (5 MIC) association. The complex was prepared as described above and then solubilized in PBS. The solution was sterilized by filtration on 0.2 μm filter and then diluted in PBS at 1000 mg/L.

Likewise, the Dsp B-CD3 and Dsp B-CD3:CIP-Ad conjugate conjugates was tested (final concentration 30 μg/mL), alone or associated with CIP-Ad. Finally, the antibiofilm activity of the Dsp B-CD3-:CIP-Ad conjugate was tested (final concentration, 30 μg/mL).

Bradford protein assay: Twenty μL of protein sample was added to 1 mL of a Bradford solution (BioRad Protein Assay) diluted (1/5, *v*/*v*). After 30 min of reaction, the absorbance was read at 595 nm and the protein concentration was determined using a calibration curve realized with a range concentration of Bovine Serum Albumin (BSA).

## 4. Discussion

Complexation assays between the adamantyl group grafted on ciprofloxacin, CIP-Ad, and four β-cyclodextrins derivatives, i.e., β-CD, 2,3-*O*-dimethyl-β-CD, 2,6-*O*-dimethyl-β-CD, and 2,3,6-*O*-trimethyl-β-CD, were investigated. A phase solubility study confirmed an increase in the CIP-Ad solubility with CD concentration, pointing out a complex formation. NMR and ITC experiments showed a stoichiometry of 1/1 of the β-CD/CIP-Ad complexes with an affinity depending on the type of methylation of β-CD. Due to their moderate affinity constant, 2,3-*O*-dimethyl-β-CD/CIP-Ad and permethylated-β-CDs/CIP-Ad complexes were selected for the antibacterial studies. The evaluation of the antibacterial activity of both CIP-Ad and β-CD derivatives/CIP-Ad complexes was performed on different *S. epidermidis* strains. MIC values showed that the 2,3-*O*-dimethyl-β-CDs did not alter the CIP-Ad activity, while complexation with permethylated-β-CDs increased MIC values. Antibiofilm assays, then performed on *S. epidermidis* biofilms, confirmed the synergistic effect observed with the association of DspB/CIP, which was partly maintained in the presence of the complex formed between the modified antibiotic CIP-Ad and the 2,3-*O*-dimethyl-β-CD. DspB was then modified with different CD conjugates through chemical covalent coupling with carboxylic permethylated CD derivatives with various spacer lengths. We showed that the modified DspB with the CD conjugates maintained their antibiofilm activities. We validated our approach by demonstrating that the DspB-permethylated-β-CD/ciprofloxacin-Ad drug delivery system exhibited an efficient antibiofilm activity. Biofilm-associated infections are particularly problematic because sessile bacteria are able to withstand the host immune defense response and are drastically more resistant to antimicrobials than their planktonic counterparts [4,73].

This is all the true in the context of implanted-medical-device- (for example, prostheses or catheters), nosocomial-, or depressed-immune-system-related infections. Among the *Staphylococcal* species that appear in the list of leading etiologic agents, *S. epidermidis* is the second after *Staphylococcus aureus* [1,2,3,73], hence our choice. In the last decade, several innovative strategies have been published to fight against biofilms, some of them being patented. Among these strategies, modifications of surfaces to prevent adhesion and/or kill adherent bacteria have been proposed [74,75]. However, these approaches are not completely satisfactory, due to their limited efficiency. Consequently, the main anti-infection tool used today in the medical field remains the antibiotherapy approach with often highly limited efficiency due to the high resistance of sessile bacteria to most antibiotics [4].

To conclude, and in light of this assessment, anti-biofilm-enzyme-based approaches (as the drug delivery system presented here) constitute a promising alternative. Due to their ability to degrade the biofilm polymer matrix, they significantly increase the efficiency of some antibiotics by both modifying the environmental conditions within the biofilm (for example, dissolved oxygen concentration levels) and by increasing drug diffusivity. Some of these enzymes are already marketed. Thus, the pulmozyme^®^ (dornase alpha), a recombinant human deoxyribonuclease I (rhDNase), is inhaled to improve lung function in cystic fibrosis [76] and mechanically ventilated patients [77]. However, as already mentioned, the main drawback of this enzymatic antibiofilm strategy is the risk of live bacteria dispersal, hence the use of antibiotics in association. However, future investigations are required in order to improve the efficacy of the approach, Finally, the present concept presents the high advantage of being an “all-in-one” tool and constitutes real progress compared to conventional antibiotherapy treatments used by clinicians today.

## Data Availability

Not applicable.

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
