# Peer review of "Modification of Dispersin B with Cyclodextrin-Ciprofloxacin Derivatives for Treating Staphylococcal"

_molecules, 2023, doi:10.3390/molecules28145311_

Round 1

Reviewer 1 Report

1.improve the introduction part

2.improve the discussion vivdly

3.add recent references instead of old reference 

4.author should be add biofilm formation and inhibition images of CV staining

5.author should be add asterisks value of each bar diagram images 

6.author should be add followed the statistical analysis 

NA

Author Response

Dear Sir 

kindly receive our response on your report.

best regards

Reviewer 2 Report

The antibiofilm of Dispersin B-permethylated-β-cyclodextrin/ciprofloxacin adamantly (DspB-β-CD/CIP-Ad). The submission can be accepted after revision considering the following points:-

 1.      The title should be revised to be clear and informative. Redundant words such as ‘vector based on a modified’;’ and ‘Complexation study and biological activity.’ Should be removed. The other component in the complex should be included in the title.

2.      ‘Drug delivery’ should be used instead of ‘vector’. ‘Vector (epidemiology), an agent that carries and transmits an infectious pathogen into another living organism.

3.      FT-IR  data confirming the complex and the interactions within the materials should be included.

4.      Abbreviations such as ‘ITC’, ‘ MIC’, … should be fully defined when mentioned for the first time.

5.      A comparison with previously published biopolymer-based biofilm should be discussed and summarized in a Table.

6.      References for biopolymers-based biofilms should be updated including these References; Colloids and Surfaces B: Biointerfaces 2015, 127, 281-291; Materials Science and Engineering: C 2019, 94, 484-492; Research in Veterinary Science 2021, 137, 262-273.. The advantages and disadvantages compared to other biopolymers should be discussed taking into consideration these References.

7.      The language should be revised and typos should be corrected.

Minors

8.      Remove terms such as ‘nano vector’.

9.      Improve References style, e.f. ‘[6][7][8][9][10][11][12].’ can be ‘[6-12].’; ‘[41][42][43][44][45]’ to be ‘[41-45]’

The antibiofilm of Dispersin B-permethylated-β-cyclodextrin/ciprofloxacin adamantly (DspB-β-CD/CIP-Ad). The submission can be accepted after revision considering the following points:-

1.      The title should be revised to be clear and informative. Redundant words such as ‘vector based on a modified’;’ and ‘Complexation study and biological activity.’ Should be removed. The other component in the complex should be included in the title.

2.      ‘Drug delivery’ should be used instead of ‘vector’. ‘Vector (epidemiology), an agent that carries and transmits an infectious pathogen into another living organism.

3.      FT-IR  data confirming the complex and the interactions within the materials should be included.

4.      Abbreviations such as ‘ITC’, ‘ MIC’, … should be fully defined when mentioned for the first time.

5.      A comparison with previously published biopolymer-based biofilm should be discussed and summarized in a Table.

6.      References for biopolymers-based biofilms should be updated including these References; Colloids and Surfaces B: Biointerfaces 2015, 127, 281-291; Materials Science and Engineering: C 2019, 94, 484-492; Research in Veterinary Science 2021, 137, 262-273.. The advantages and disadvantages compared to other biopolymers should be discussed taking into consideration these References.

7.      The language should be revised and typos should be corrected.

Minors

8.      Remove terms such as ‘nano vector’.

9.      Improve References style, e.f. ‘[6][7][8][9][10][11][12].’ can be ‘[6-12].’; ‘[41][42][43][44][45]’ to be ‘[41-45]’

Author Response

Dear sir,

kindly receive our response on your questions and comments.

best regards

Reviewer 3 Report

Title: An antibiofilm vector based on a modified β-cyclodextrin: Complexation study and biological activity

This manuscript is well-written by the authors. I do believe that if they can improve the manuscripts following all comments. It might have a chance to publish in the journal.

Comments

Topic: Please rewrite. The authors may include Staphylococcus epidermidis because the authors have focused on S. epidermidis biofilms.

Abstract

1. Line 14: Please write in passive voice.

2. Please re-write the manuscript using past tense.

3. Line 25: S. epidermidis; please write the full name “Staphylococcus epidermidis”, followed by a shot name “S. epidermidis

Introduction

4. Line 37: Staphylococcus epidermidis should be written in italic.

5. All scientific names should be written in italic.

6. Line 46: Please correct the style of references “[6][7][8][9][10][11][12]” or [6-12]

7. Introduction is too long. Please identify the problem of the study and describe why do you interest in this study.

8. Please include the objective at the end of introduction.

9. It is better if the authors include materials and methods in the manuscript.

Results and discussion

10. Some Tables or Figures should be prepared as supplementary data.

11. All the results; please describe the compact and key the results.

12. All Figures should be used the same format

13. Line 447 and 448: please check the use of Bold symbol such as [ or [, please correct in the whole manuscript.

14. Please include more information in discussion to support the results.

15. Please include conclusion

References.

16. There are many references. In general, there are 30-40 references for each manuscript (research articles). Please delete some unnecessary references. Please cite update references.

Please correct the grammar. 

Author Response

(The authors gave the same response as above.)

Round 2

Reviewer 1 Report

Accept

need to extensive edit 

Author Response

Point 1: improve the introduction part.

Response 1: Please see the correction highlighted in red in the manuscript.

Point 2: improve the discussion vivdly.

Response 2: Please see the correction highlighted in red in the manuscript.

Point 3: Author should add biofilm formation and inhibition images of CV staining.

Biofilm of S. epidermis strain RP62A treated with 0,01 µg/mL Dsp B.

Response 3:  We have some images of the biofilm of Staphylococcus epidermidis untreated and another two images of the biofilm after treatment with different concentrations of Dispersin B. We decided not to include these images in the manuscript because we didn’t do the CV staining for all the strains.

Biofilm of S. epidermis strain RP62A non treated.

.

Biofilm of S. epidermis strain RP62A treated with 0,1 µg/mL Dsp B.
